# A Method for Evaluating Robustness of Limited Sampling Strategies—Exemplified by Serum Iohexol Clearance for Determination of Measured Glomerular Filtration Rate

**DOI:** 10.3390/pharmaceutics15041073

**Published:** 2023-03-27

**Authors:** Markus Hovd, Ida Robertsen, Jean-Baptiste Woillard, Anders Åsberg

**Affiliations:** 1Section for Pharmacology and Pharmaceutical Biosciences, Department of Pharmacy, University of Oslo, P.O. Box 1068 Blindern, 0316 Oslo, Norway; ida.robertsen@farmasi.uio.no (I.R.); anders.asberg@farmasi.uio.no (A.Å.); 2Inserm, Univ. Limoges, CHU Limoges, Pharmacology & Toxicology, U 1248, F-87000 Limoges, France; jean-baptiste.woillard@unilim.fr; 3Department of Transplantation Medicine, Oslo University Hospital, P.O. Box 4950 Nydalen, 0424 Oslo, Norway

**Keywords:** limited sampling strategies, population pharmacokinetic modelling, semi-parametric simulation, robustness, therapeutic drug monitoring, area under the curve, AUC, glomerular filtration rate, GFR

## Abstract

In combination with Bayesian estimates based on a population pharmacokinetic model, limited sampling strategies (LSS) may reduce the number of samples required for individual pharmacokinetic parameter estimations. Such strategies reduce the burden when assessing the area under the concentration versus time curves (AUC) in therapeutic drug monitoring. However, it is not uncommon for the actual sample time to deviate from the optimal one. In this work, we evaluate the robustness of parameter estimations to such deviations in an LSS. A previously developed 4-point LSS for estimation of serum iohexol clearance (i.e., dose/AUC) was used to exemplify the effect of sample time deviations. Two parallel strategies were used: (a) shifting the exact sampling time by an empirical amount of time for each of the four individual sample points, and (b) introducing a random error across all sample points. The investigated iohexol LSS appeared robust to deviations from optimal sample times, both across individual and multiple sample points. The proportion of individuals with a relative error greater than 15% (P15) was 5.3% in the reference run with optimally timed sampling, which increased to a maximum of 8.3% following the introduction of random error in sample time across all four time points. We propose to apply the present method for the validation of LSS developed for clinical use.

## 1. Introduction

The area under the plasma concentration-time curve (AUC) is a clinically useful variable for systemic drug exposure. Within several therapeutic fields, AUC-targeted therapeutic drug monitoring (TDM) is becoming more clinically acknowledged [1]. Accurate estimation of AUC either requires multiple samples within a dose interval when applying the trapezoidal method, or knowledge of the individuals’ pharmacokinetic parameters, e.g., clearance. The use of the trapezoidal method in this aspect is time-consuming for both patients and healthcare professionals and not feasible in a clinical setting. However, with Bayesian estimates (BE) based on, for example, a population pharmacokinetic model or the use of a linear regression model, accurate estimates of pharmacokinetic parameters and AUC may be obtained by using a limited number of optimally timed samples [2]. Such limited sampling strategies (LSS) may reduce the number of samples and limit the length of the study visit to make AUC-targeted TDM clinically applicable [3].

In a real-life setting, it is not uncommon for actual sample times to deviate from the optimal LSS sample times. In contrast to multiple linear regression (MLR) models where coefficients are determined for pre-defined or binned sample times, BE approaches are generally considered more flexible with regard to the timing of the samples, as long as the exact sample times are recorded [4].

According to pharmacokinetic theory, clearance of an intravenously administered drug may be determined by dividing the dose by the AUC. The glomerular filtration rate (GFR) is a clinically important marker for renal function and is typically estimated from blood concentrations of endogenous markers (eGFR). However, the most accurate metric of renal function is the measured GFR (mGFR) assessed by determining the AUC of an exogenous substance subject to clearance via filtration in the kidney [5]. The gold standard of these exogenous markers is inulin but it is difficult to obtain injection-quality inulin nowadays and the analytical assay is also somewhat challenging. Due to this, the contrast agent iohexol has become the new gold standard for mGFR as it shows high concordance with inulin-derived mGFR given optimal sampling times in relation to absolute GFR level [6]. Iohexol exhibits a low degree of protein-binding, low toxicity for the needed doses, no tubular secretion or reabsorption, and is generally stable in plasma/serum [7]. As iohexol is fully excreted by the kidneys, mGFR may be determined by measuring the clearance of iohexol. For this, both MLR- and BE-based LSS are available in the literature. Of these, BE-based methods have been shown to be more flexible and accurate than MLR [8].

We have previously demonstrated the feasibility of a BE-based 4-point LSS to accurately determine mGFR over the range of 14 to 149 mL/min using iohexol serum clearance [9]. Our LSS includes four samples within 5 h following intravenous administration of iohexol. Here, we accurately determine the iohexol serum clearance by dividing the administered dose by the AUC. As such, this method is equally viable for evaluating the effect of shifts in sample time on AUC, as well as mGFR. While the effect of a deviation in time from LSS based on MLR has been evaluated previously [10], the effect of deviations in sample time on parameter estimates in the BE-based methods has not been readily studied and is rarely considered during LSS development or their clinical use. In this work, we demonstrate a general method for evaluating the robustness of an LSS, using the iohexol model for AUC-based mGFR determination as an example. The effect of deviations in time, both across individual and multiple time points, on AUC and model estimated parameters are evaluated.

## 2. Materials and Methods

### 2.1. Population Pharmacokinetics Model and Limited Sampling Strategy of Iohexol

The population pharmacokinetic model and associated LSS for iohexol serum clearance have previously been described in detail [9]. In short, a non-parametric adaptive grid (NPAG) approach implemented in Pmetrics [11] for R [12] was used. The model consisted of two compartments, parameterized in clearance (CL) from the central compartment, the volume of central (V) and peripheral (Vp) compartments, and inter-compartmental blood flow (Q), allometrically scaled for body weight using power factors of 0.75 for CL and Q and 1 for V and Vp. The model was developed on rich data from 176 patients (1131 samples), and externally validated in a cohort of 43 patients (395 samples). The 4-point sampling strategy optimized for clinical use included samples at 10 min, 30 min, 2 h, and 5 h following intravenous administration of 3235 mg iohexol (Omnipaque 300 mg I/mL, GE Healthcare AS, Oslo, Norway). A public, web-based interface to this model was developed and is freely available at https://www.mgfr.no.

### 2.2. Semi-Parametric Simulation from Support Points

To evaluate the robustness of the previously developed LSS of iohexol, simulations were performed to obtain pharmacokinetic profiles from a similar parameter distribution (i.e., population) as the original dataset. Our simulation method did not include covariates, and as such, a covariate-free version of the model was used. This model was developed and evaluated using the original development and validation datasets. Model diagnostic plots and performance metrics are available in Appendix A.

Briefly, the NPAG algorithm estimates the joint population parameter distribution, which is used as a Bayesian prior for individual parameter estimation. The algorithm has recently been explained in detail by Yamada et al., 2020 [13]. The population parameter distribution is a discrete distribution provided as a set of support points, each a vector of length *D* with an associated probability, where *D* is the number of parameters. The discrete distribution may be transformed to a continuous distribution for the purpose of sampling a wider range of possible parameter combinations. To accomplish this, we assume a Gaussian distribution over each support point, forming a Gaussian mixture distribution. The probability density function for the multivariate Gaussian mixture is defined as
(1)$$p\left(x\;|\;\mu ,\Sigma \right)=N\left(\mu ,\;\Sigma \right)$$
where *μ* and *Σ* are the vector of means and the matrix of variances, respectively. Values of μ are readily obtained from the individual support point vectors. In order to determine *Σ*, the univariate Gaussian mixture was evaluated for each parameter, the density for which is
(2)$$p\left(x\right)=\sum_{i=1}^{K}\pi_{i}*N(x\;|\;\mu_{i},\;\sigma_{i})\;\;\;\;\mathrm{satisfying}\;\;\;\;\sum_{i=1}^{K}\pi_{i}=1$$
where *π* is the weighting (or probability) for the *K*th Gaussian distribution with mean *μ* and variance *σ*. For each parameter, a common *σ*, and thus, the proposed element of *Σ*, was determined by minimizing the sum of the squared distance between the simulated and observed (individual posterior) parameter distribution. Minimization was performed using the built-in optim-function in R, implementing Brent’s method. Sampling from the mixture distribution is achieved by first sampling the mixture components, i.e., the support points, with replacement, weighted by their probability. Then, multivariate normal sampling of parameters was accomplished using the rtmvnorm function implemented in the tmvtnorm (version 1.5) package for R (version 4.1.3) [14]. Rejection sampling was used to respect the boundaries of the population pharmacokinetic model. A successful simulation was evaluated by the overlapping index for empirical distributions [15], for which values equal to or above 85% were considered acceptable, comparable to an error of 15%. In order to generate concentration-time profiles from the simulated parameter vectors, the population pharmacokinetic model was rewritten to be used in the mrgsolve [16] package for R (Appendix A). Simulated sampling was performed in 1 min intervals from 0 to 24 h following a dose of 3235 mg of iohexol. No systematic or random error was added to the measurements. 

GFR was calculated by dividing dose by the AUC from zero to infinity (AUC_0-∞_). Simulated profiles with GFR < 15 mL/min or GFR > 115 mL/min were excluded, as they were outside the validated range of the LSS and will not be explored in this work. As such, both AUC and GFR are conversely evaluated in this work.

### 2.3. Deviation from Optimal Sample Times

The robustness of the 4-point sampling strategy for iohexol serum clearance was evaluated at each of the sample points with empirically selected deviations in time; 10 min (±2, 4, 5, and 6 min), 30 min (±5, 10, and 15 min), 2 h (±5, 15, 30, and 60 min), and 5 h (±5, 15, 30, 60, 120 and +180, 420, and 1140 min), in addition to a reference run with the original sample times. Each shift was run separately, with cycling, and using the support points of the covariate-free model ran on the complete dataset as a Bayesian prior, as specified in the original publication [9].

In order to evaluate the effect of deviation over multiple sample times, a random normally distributed error, centered around each respective sample point, and with a relative standard deviation (RSD) of 5, 10, 15, 20, and 25% was added to all sample points, truncated (using rejection sampling) at each point to prevent overlap; 10 min (5–15 min), 30 min (15–60 min), 2 h (1–3 h), and 5 h (3–8 h). As a measure of robustness to the aforementioned shifts in sample times, both the mean absolute prediction error in mGFR and the proportion of individuals with relative prediction error greater than 15% (P15) were used. Here, a P15 less than 15% was considered acceptable.

### 2.4. Optimal Sample Windows

Based on the results of deviation in both individual and multiple sample points, two approaches to empirical sample windows were used. For deviation in individual sample time, assuming otherwise no deviation in the remaining sample points, the time intervals for which the mean error was lower than 2 mL/min may be used. For deviation across all sample times, the level of RSD associated with an acceptable P15 was used to calculate empirical sample windows for all sample points by calculating the 90% confidence interval for the normal distribution centered at each sample point, truncated to avoid overlap between samples.

## 3. Results

### 3.1. Simulated Profiles

A total of 400 pharmacokinetic profiles were simulated, of which 58 and 3 were excluded due to a simulated GFR of less than 15 mL/min or greater than 115 mL/min, respectively, yielding a total of 339 profiles used in the analysis. The variances that minimized the distance between observed and simulated parameter densities were 0.61, 2.2, 1.2, and 0.9 units, for CL, Q, V, and Vp, respectively. Simulated parameter densities demonstrated satisfactory overlap with the observed posterior parameter densities from the original population pharmacokinetic model (91, 92, 90, and 86% for CL, Q, V, and Vp, respectively) (Figure 1). Compared to the posterior, none of the simulated parameters had a difference in weighted mean greater than 15% (Table 1). The simulated profiles (*n* = 339) were further grouped based on the estimated mGFR in relation to the chronic kidney disease (CKD) stages; stage 4:15–29 mL/min (*n* = 90), stage 3B: 30–44 mL/min (*n* = 100), stage 3A: 45–59 mL/min (*n* = 57), stage 2: 60–90 mL/min (*n* = 73), and stage 1: >90 mL/min (*n* = 19).

### 3.2. LSS Performance on Simulated Profiles

The LSS performance on the simulated profiles was evaluated by sampling at precisely 10 min, 30 min, 2 h, and 5 h. The mean absolute and relative error in GFR were 1.5 ± 2.2 mL/min and 4.1 ± 5.5%, respectively (Table 2). In total, 6.5% of the simulated profiles demonstrated an absolute error greater than 5 mL/min, and 1.2% demonstrated an error greater than 10 mL/min. The proportion of individuals with an error larger than 15% (P15) was 5.3%, seemingly increasing with decreased GFR, as expected (Table 2).

### 3.3. Effect of Shifts in Sample Times on Estimated GFR

A graphical representation of the effect of deviations in individual sample time on estimated GFR is shown in Figure 2. In all cases, the mean absolute error was below 4 mL/min, and the median absolute error was below 2.5 mL/min. For the 10 min sample, delays by up to 6 min increased P15 to a maximum of 9%. Sampling 5 and 6 min prior, effectively at 4 and 5 min post-dose, was not evaluable by the model, and these times were not included. In contrast, delaying the 30 min sample less than 15 min reduced P15 to 4%, while sampling up to 15 min earlier increased P15 to 16%. The 2 h sample exhibits a similar pattern, with reduced P15 for delayed samples, down to 4% at 60 min delayed. The 5 h sample was mostly unaffected by up to 7 h delay in sampling. However, delaying the sample to 24 h post iohexol administration drastically reduced the predictive performance as expected; P15 increased to 18%, and the mean absolute error was 2.9 ± 3.5 mL/min. In order to evaluate these trends for each CKD stage, the median error for each shift is shown in Figure 3.

Applying random, normally distributed noise with an RSD equal to 5 and 10% across all sample times led to a P15 of 7.7% in both cases, and a maximum P15 of 8.3% was achieved in the case of an RSD of both 20% and 25% (Figure 4). For the simulated profiles with CKD stage 1 (GFR 90–115 mL/min), P15 was 0% for all levels of RSD, and a maximum of 4.1% at 5% RSD for profiles with GFR 60–90 mL/min. In contrast, simulated profiles with GFR between 45–55 mL/min, 30–44 mL/min, and 15–15 mL/min incurred a P15 of 7%, 14%, and 10% at 25% RSD, respectively.

### 3.4. Optimal Sample Windows

The intervals around the deviation in individual optimal sample times that provide a mean error in predicted GFR less than 2 mL/min, and conversely a low error in AUC, were 6–16 min (10 min), 20–45 min (30 min), 1.5–3 h (2 h), and 4.75–12 (5 h) (Figure 5A). A careful estimate of the optimal sample window may be obtained by, e.g., the 90% confidence interval for the normal distribution with optimal sample times as the mean, and an RSD of 25%. As such, an estimate of optimal sample windows for the present LSS, without considering the absolute renal function of the patient, are 6–12 min (10 min), 18–42 min (30 min), 1–3 h (2 h), and 2.5–7.4 h (5 h) (Figure 5B), assuming no overlap.

### 3.5. Effect of Shifts in Sample Times on Model Parameters

Supplementary to the effect of a deviation from optimal sample time on predicted GFR, and conversely, predicted AUC_0-inf_, changes in estimated model parameters were also evaluated. A graphical representation of the model-estimated parameter densities across all evaluated shifts is shown in Figure 6. When compared with the true parameter densities of the simulated data, the reference run with optimally timed samples achieved a relative error in mean population model parameter estimates in CL, V, Vp, and Q of 1.5%, 0.6%, 6.9%, and 3.8%, respectively. However, the mean individual relative errors in the same estimates in CL, V, Vp, and Q were 7.4%, 7.6%, 22.7%, and 495%.

## 4. Discussion

In this work, we demonstrate an intuitive approach to evaluating the robustness of LSS, with direct clinical applications. The method was applied to a previously published model for serum iohexol clearance used for the determination of mGFR based on accurate estimates of AUC_0-inf_. To our knowledge, this is the first work evaluating the robustness of LSS in such a setting. Overall, the 4-point LSS appears robust to shifts in both single and multiple sample times, especially for profiles with medium to good GFR, i.e., above 45 mL/min. An interesting finding is that the robustness is affected by patient absolute clearance or GFR, in this case, and that acceptable sample time deviations should be adapted also based on this information. This is especially useful in scenarios when a rough estimate of the patient clearance is known based on clinical history, but the exact mGFR is desired, e.g., for dose-adjustment of drugs.

In the case of LSS employing MLR, Sarem and colleagues have previously evaluated the effect of deviations in sample time on AUC only [10]. The present work applies this methodology to BE-based LSS and evaluates not only the effect of such deviations on AUC but also the effect on parameter estimates at the individual and population levels. Additionally, BE-based methods have been shown to outperform LSS based on MLR in the case of iohexol clearance for the determination of mGFR [8,9]. The BE-based method was evaluated with a restriction of sampling within standard laboratory opening hours, i.e., the whole procedure was finalized within 5 h, while the MLR-base method allowed sampling up to 24 h after dosing. The BE-based model was not only more accurate but also better adapted to clinical practice [9]. With the development of easily and freely accessible interfaces to these otherwise complicated BE-based models, such as the one we provide at https://www.mgfr.no, the barrier to implementation in a clinical setting is significantly lowered, becoming similar to that of MLR.

When evaluating deviations in sample time for individual sample points, and assuming otherwise optimal sampling, no clinically significant increase in either mean absolute error or P15 was found across shifts in the 10 min sample, and the 30 min sample may be delayed by 15 min, even favorably so. As for the 2 h and 5 h samples, either may be accelerated by up to 30 min without increasing the P15 above 10%. This may potentially save time for both the patient and healthcare personnel during AUC-guided TDM. In this work, delaying individual samples improved the predictive performance of the LSS, likely due to the abundance of simulated profiles with low GFR. Previous history or indication of the patient’s AUC and/or mGFR, it is possible to make specific recommendations. For example, parameter estimation in patients with high AUC, and conversely, low mGFR, may benefit more from delayed sampling, and vice versa. A challenge is that individual pharmacokinetic parameters are subject to change over time, but a Bayesian framework compatible with the present method has previously been described by Bayard and Jelliffe [17].

The introduction of random error with an RSD of 25% was associated with a P15 of 8.3%, compared to 5.3% in the reference run. The level of RSD yielding an acceptable P15 could be viewed as a surrogate marker for LSS robustness to shifts across multiple sample times for implementation in the clinic. As demonstrated, this may be tailored to the study population as a whole, or individual sub-groups of patients with, e.g., different stages of CKD. The empirically determined optimal sample windows allow for added granularity with regard to the diligence required for sample collection. However, this does not address the minimum distance required between two given samples, which is likely to affect the accuracy of parameter estimates.

With respect to the model estimated parameters, the effect of empirical deviations in individual sample times on the population level was negligible, as indicated by a low relative error in mean parameter values and the fact that the population parameter densities mostly overlapped the simulated, true density. However, individual parameter estimates varied significantly—especially the peripheral volume and inter-compartmental clearance were often misidentified. This is not surprising, as these parameters are seldom identifiable. This misidentification did not have any effect on the predictive performance of the model, i.e., estimates of individual AUC_0-∞_, here translated to mGFR. All runs exhibited exceptionally low mean prediction errors and relative root-mean-squared errors. This was observed during early method development and for this reason, iohexol serum clearance, and thus GFR, was calculated by dividing dose by AUC_0-∞_. This further highlights the need for a more robust evaluation of LSS, especially when model parameters are used directly. Our results demonstrate the clinical application of evaluating the robustness of BE-based LSS. Previously, the effect of a deviation in sample time was unknown but has now been quantified for the present model and population. With this information, one may look up the deviation in sample time for the relevant CKD stage and use this to decide on whether to include an additional sample, for example, which is likely to improve the accuracy of the parameter estimates. Such changes to the LSS are not possible in the case of multiple linear regression-based methods, where one is restricted to a pre-defined or binned sample space.

For simulation-based studies, it is imperative that the simulated population reflects the underlying research question. While this is implicitly assumed, it is not usually confirmed in simulation-based studies, despite its importance. In this work, we aimed to simulate profiles from a similar population, which was confirmed by evaluating the overlap in parameter densities, in addition to comparisons of weighted mean and median. A disadvantage of the proposed method for semi-parametric simulation is the lack of covariates, given that multiple model parameters were allometrically scaled in the original model. Our strategy for simulation was based on the mechanistic interpretation of the support points representing the discrete population parameter distribution from the NPAG algorithm. However, there is no direct link between the support points and the covariates. For the covariate to be included in the multivariate normal sampling, a sensible mean and variance must be provided. An initial choice would be the observed mean and variance of the covariate, which was attempted during method development, but led to poor overlap between the posterior and simulated parameter densities. Alternate approaches to include covariates in a semi-parametric simulation will be investigated in a future work. Even though the present work utilized a covariate-free version of the original model, we still believe that the proposed simulation method provides an accurate representation of the effect of deviations in LSS sample times on individual pharmacokinetic estimates, as the method for evaluating the robustness of LSS is agnostic to the process for which data is generated.

## 5. Conclusions

By deviating from the optimally timed sample point(s) of an LSS either empirically or randomly, the robustness of the LSS to such shifts can be approximated. Additionally, empirical optimal sample windows may be obtained for a more flexible sampling schedule. It was further revealed that despite model population parameter estimates being within 10% across all evaluated deviations, individual model parameter estimates were prone to misidentification. These findings provide additional insight into the necessary diligence required during sample collection of optimally timed samples and provide a method for evaluating LSS robustness with respect to both pharmacokinetic (i.e., AUC) and model estimated parameters. We propose the present method be applied during the development and validation of LSS for clinical use.

## Figures and Tables

**Figure 1 pharmaceutics-15-01073-f001:**
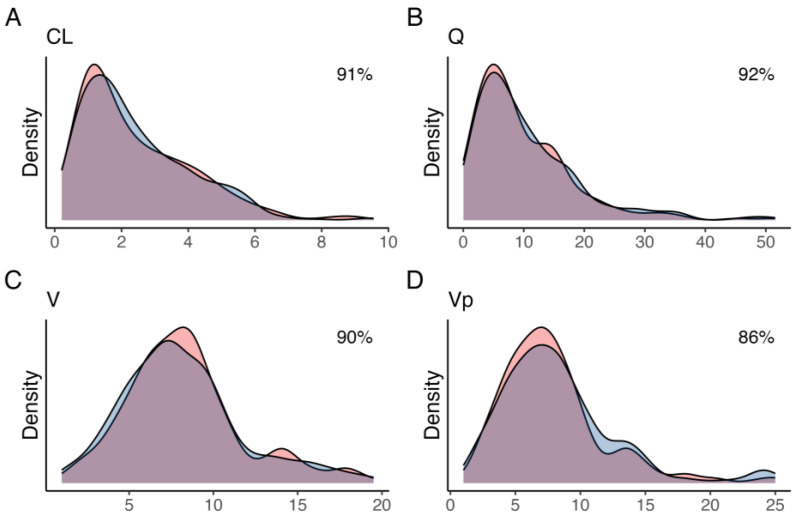
Kernel density estimates for the posterior (blue) and simulated (red) for (**A**) clearance from the central compartment (CL), (**B**) inter-compartmental clearance (Q), (**C**) central volume (V), and (**D**) peripheral volume (Vp). The overlap coefficient for empirical distributions between the posterior and simulated parameter distribution is shown in the top-right corner.

**Figure 2 pharmaceutics-15-01073-f002:**
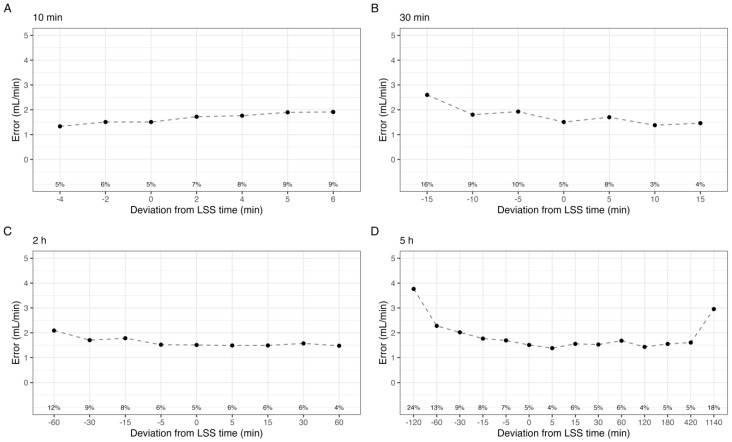
Mean absolute error in estimated GFR by the shift in sample time at point (**A**) 10 min, (**B**) 30 min, (**C**) 2 h, and (**D**) 5 h. Labels indicate the percentage of individuals with a relative error greater than 15% (P15).

**Figure 3 pharmaceutics-15-01073-f003:**
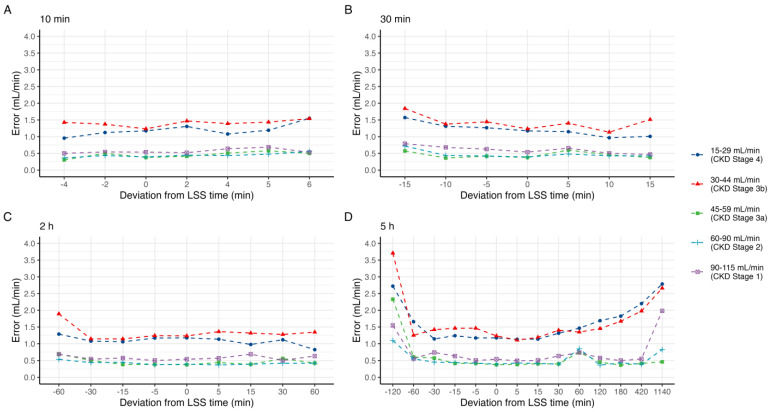
Median absolute error in predicted mGFR by deviation in sample time at point (**A**) 10 min, (**B**) 30 min, (**C**) 2 h, and (**D**) 5 h, grouped by CKD stage.

**Figure 4 pharmaceutics-15-01073-f004:**
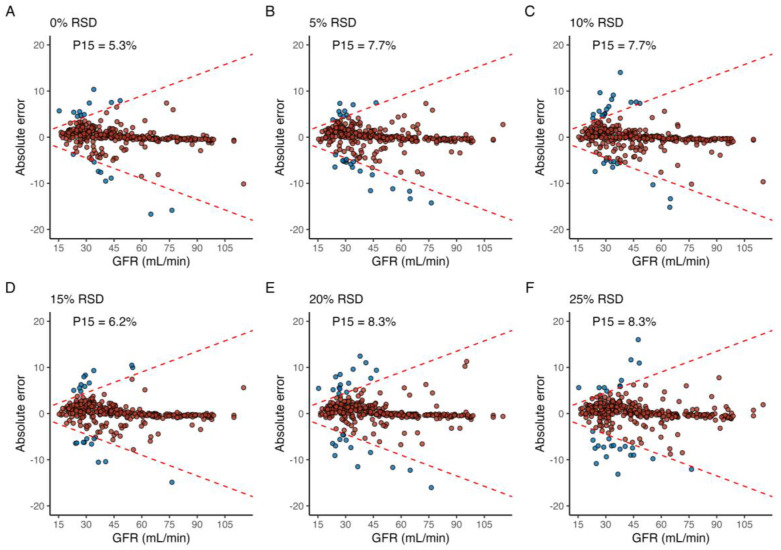
The effect of shifts in time over multiple sample points, where the (**A**) reference run is compared to when shifts in sample time are normally distributed around the optimal sample time with a relative standard deviation (RSD) of (**B**) 5%, (**C**) 10%, (**D**) 15%, (**E**) 20%, and (**F**) 25%. Blue and red fill indicates an individual error greater than or less than 15%, respectively. The label in the upper-left corner denotes the proportion of individuals with a relative prediction error greater than 15% (P15) for each level of RSD.

**Figure 5 pharmaceutics-15-01073-f005:**
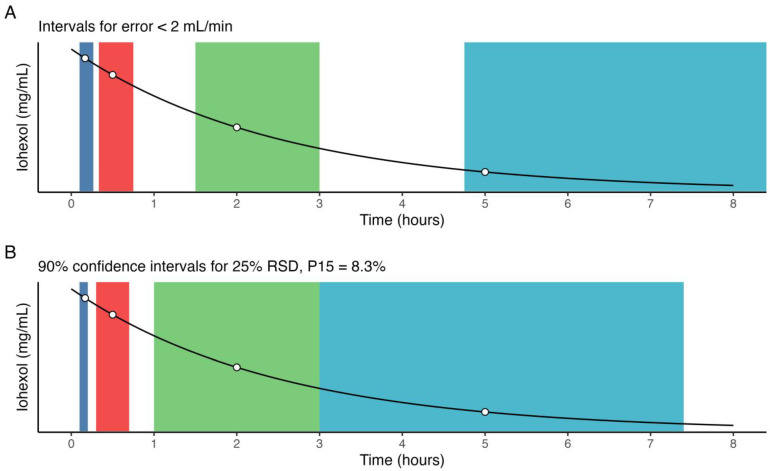
Empirical estimates of optimal sample windows for (**A**) deviations in individual sample times, assuming all other points are sampled optimally, resulting in mean error < 2 mL/min, and (**B**) deviations across all sample times, normally distributed around each sample point with 25% RSD, resulting in a P15 = 8.3%.

**Figure 6 pharmaceutics-15-01073-f006:**
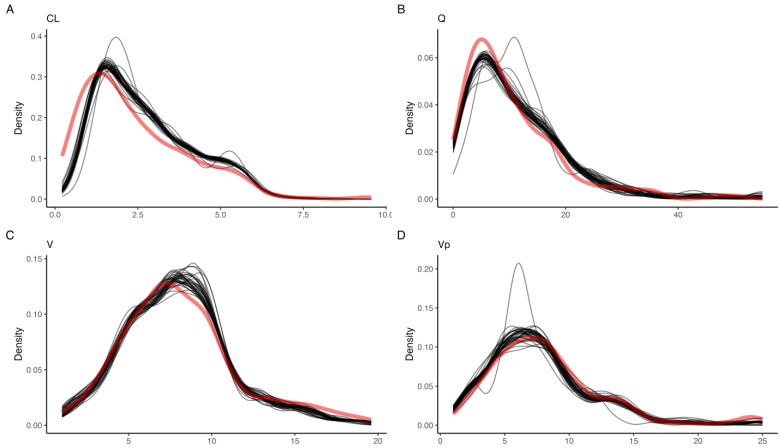
Kernel density estimates for all empirical deviations in time (black), compared to the observed posterior (red) for (**A**) clearance from the central compartment (CL), (**B**) inter-compartmental clearance (Q), (**C**) central volume (V), and (**D**) peripheral volume (Vp).

**Table 1 pharmaceutics-15-01073-t001:** Weighted mean and weighted median (95% credibility interval) of the population pharmacokinetic model parameters support points for the original and simulated dataset.

	Weighted Mean		Weighted Median (95% Credibility Interval)	
	Original	Simulated	Original	Simulated
CL (L/h)	2.89	2.84	1.95 (1.54–2.60)	2.42 (2.16–2.72)
V (L)	10.36	9.32	10.11 (9.19–10.91)	8.98 (8.25–9.57)
Vp (L)	9.20	7.98	7.95 (7.23–8.60)	7.46 (7.06–7.81)
Q (L/h)	10.65	11.37	8.03 (6.53–9.23)	8.65 (7.50–9.72)

**Table 2 pharmaceutics-15-01073-t002:** Limited sampling strategy performance on determining mGFR for the simulated profiles, presented as the absolute and relative error from the simulated “true” GFR. Data are presented as mean ± standard deviation.

Group	Absolute Error (mL/min)	Relative Error (%)	P15 (%)	*n*
All profiles	1.5 ± 2.2	4.1 ± 5.5	5.3	339
CKD Stage 4 (15–29 mL/min)	1.5 ± 1.3	6.3 ± 5.8	7.8	90
CKD Stage 3b (30–44 mL/min)	1.9 ± 2.2	5.4 ± 6.0	8.0	100
CKD Stage 3a (45–59 mL/min)	1.0 ± 1.6	2.2 ± 3.4	1.8	57
CKD Stage 2 (60–90 mL/min)	1.3 ± 3.0	1.9 ± 4.4	2.7	73
CKD Stage 1 (90–115 mL/min)	1.2 ± 2.2	1.2 ± 1.9	0.0	19

## Data Availability

The data presented in this study are available on request from the corresponding author.

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
