# Peer review of "A Method for Evaluating Robustness of Limited Sampling Strategies—Exemplified by Serum Iohexol Clearance for Determination of Measured Glomerular Filtration Rate"

_pharmaceutics, 2023, doi:10.3390/pharmaceutics15041073_

Round 1

Reviewer 1 Report

This manuscript describes an approach for evaluating the robustness of a limited sampling strategy using estimation of GFR by iohexol clearance as example. The presented simulation study is based on a previously reported population pharmacokinetic model by the last author. The manuscript is well written, results comprehensiveley presented, and can clinically direclty be applied for GFR estimation in a freely accesible web-tool. I only have minor suggestions.

Methods: if possible, a schematic illustration of the modeling - simulation - evaluation workflow would be helpful to follow the different steps described.

Discussion: it would be interesting if the proposed approach for developing and evaluating an optimal/limited sampling strategy could be contrasted to other approaches proposed in the literature, also to understand novelty of the presented approach. In total only 12 references are cited which is probably referring only to a subset of relevant papers in this area.

Author Response

Please see the attachment for the complete point-by-point replies.

Reviewer 2 Report

The manuscript presents the possibility to optimize limited sampling strategies in population pharmacokinetic modeling. Glomerular filtration rate was calculated on the basis of iohexol clearance. On the basis of its clearance, the effect of sample time deviations was evaluated. The approach allows evaluation of the robustness of limited sampling strategies. The applied NPAG algorithm helped to avoid some of the associated maximum-likelihood problems. The study has significant value for the practice in relation to good practices in the therapeutic drug monitoring. The manuscript is well written and it can be accepted for publication.

Introduction

This section is well written and clearly introduces the topic. It would be of help for the readers if in one or two sentences the advantage of determination of iohexol clearance is summarized.

Material and methods

This section contains enough clear explanation about the steps of development of the population model and the simulations based on this model. The cited articles are adequate and contain the information about the methodology of modelling in detail.

Results

The results are well explained and depicted. The presentation of the data as mean a median and range in Table 1 brings additional value which helps for evaluation of the model. Figures are representative and clear.

The place of the Figure 2 has to be adjusted so that it should be on one page with its heading.

The results show the ability of the authors to present in understandable way not very easy mathematical modelling and the significance of their results for the clinics.

Figure 6: Please, indicate in the heading of the figure where the parameters were shown (as in Figure 1).

Discussion

Discussion and conclusion reflect accurately the results. The discussion explained not only the advantages but also the limitations of the study.

Author Response

(The authors gave the same response as above.)

Reviewer 3 Report

In their study, Hovd et al used a 4-point LSS to evaluate the impact of deviation from actual optimum sampling time point. The underlying concept is not new and many LSS model using population PK and Bayesian approach have been developed over the years for many drugs and the deviation from actual sampling time point(s) have been discussed.  There are several caveats with Hovd et al's LSS model.  

Limited sampling models (LSM) or limited sampling strategy (LSS) uses primarily one blood sample or two for highly variable drugs. The LSM models are based on linear regression and have been found to be fairly accurate and of practical value in the clinical settings. The trick in LSM or LSS is that the optimal sampling time is the best in the terminal phase.  In other words, for Iohexol the 8-hour sampling time is probably the most ideal optimal sampling. A 5-hour sampling time may be also suitable but will carry slightly higher prediction error than 8-hour sample.

Hovd et al used a 4-point LSS which should not be called LSS because there are too many time points in this model.  On top of it, the authors used a population PK-based Bayesian method. This is basically taking a simple thing and complicating it. Iohexol has a half-life of  2 hours and the number of blood samples (extensive sampling) for a compound with such a short half-life probably will be 5-6.  With 4 time points for such a short half-life compound one can simply use a non-compartmental analysis to estimate AUC or other relevant PK parameters.  One also does not need to search optimal sampling time points with 4 samples. For Iohexol, two time points in the early phase and two time points in the terminal phase will give an accurate AUC value compared to 5-6 blood samples and can be very easily estimated by non-compartmental analysis.  

In reality, one can estimate Iohexol AUC from one or two blood samples using simple linear regression. Taubert et al (Taubert Max. Advancement of pharmacokinetic models of iohexol in patients aged 70 years or older with impaired kidney function.  Nature. Scientific Reports | (2021) 11:22656. www.nature.com/scientificreports) state "The feasibility of limited sampling strategies with one to three plasma samples has been evaluated in numerous studies (24, 25 references in Taubert et al's manuscript). Overall, two or even a single plasma sample was found to be sufficient for obtaining precise clearance estimates (reference 7 in Taubert et al's manuscript ). Previous studies suggested blood samples at 2 and 4 h post injection, with delayed sampling after 5 to 8 h post injection in CKD patients (25). In contrast, we found that the combination of an early sample (e.g., 30 min post injection) with a sample obtained after 300 min appears to be sufficient to precisely estimate iohexol clearance. The shortened post injection period is an important and clinically relevant advantage for implementing iohexol measurement in routine diagnostics.

Gaspri et al (Glomerular filtration rate determined from a single plasma sample after intravenous iohexol injection: is it reliable?  J Am Soc Nephrol  1996;7:2689-93) compared one-time point LSS with 6-time point Iohexol clearance to determine GFR. The optimal time for sampling was 10 h (40 mL/min/1.73 m2), 4 h (40-99 mL/min/1.73 m2), 3 h (>100 mL/min/1.73 m2). A good correlation was noted (Y = 0.968X + 1.704, r2 = 0.988) between plasma concentrations and the optimal sampling time point. Results of the study showed that for 75% of the patients, the simplified technique gave an error between -5% to +5% in the evaluation of GFR; for the remaining 25% of the patients, prediction error ranged from -22% to +40%. The authors rejected the single-point method because " the regression intercept was statistically different from 0 and the standard error of the slope estimate established that 95% confidence interval did not include 1.0 (the line of identity), thus indicating that the model can be rejected by the data at a significance level of 0.05".         

Gaspri et al's conclusion based on pure statistics is flawed.  In fact, the model provided excellent results despite the fact that the authors' model lacked the 'center specificity'. Results of Gaspri et al study indicated that for 75% of the patients, one-point LSM gave an error between -5% to +5% in the evaluation of GFR; for the remaining 25% of the patients, prediction error ranged from -22% to +40%.  The fact that 75% subjects were within 5% prediction error indicate that the model was very robust. Generally, a 10-15% prediction error in an individual subject is acceptable.  Had the authors' chosen a more realistic prediction error and considered more from practical perspective rather than relying on some statistics they would have accepted the model. For a LSM, root mean square error (RMSE) and mean absolute error (MAE) 15-20% and 10%, respectively, is acceptable.

I mentioned the above two examples because this entire exercise is very simple. Hovd et al should use one sample at 8 hour (if available) and using a regression analysis predict the AUC of Iohexol to approximate GFR in an individual.  This will be of practical value. They can also use some random deviation time around the optimal time point and evaluate how far off from the optimal sampling time the prediction starts getting worse.  My guess is that at least 30 minutes on either direction will be OK (8-hr sampling).

At the moment, Hovd et al's work has of little practical value in clinical settings. The only parameter needed is AUC or CL (dose/AUC) in the clinical setting to approximate GFR in an individual. Rest of the PK parameters such as V, Vp and Q are not needed.

The bottom line here is that more than three-decades of experience with the LSM has taught us that one or two blood sample(s) taken at some optimal time and using linear regression analysis is robust enough to estimate AUC and ultimately dose adjustment for an individual.  Off course, one can use complicated models like this one with too many blood samples which make the model not applicable in clinical settings. 

Author Response

(The authors gave the same response as above.)

Reviewer 4 Report

The authors evaluated the robustness of LSS, a limited sampling strategy, with iohexol, that has been used to measure GFR in clinics. Overall, the manuscript is well-written and clearly demonstrates that the LSS for iohexol provides a good PK estimate, especially CL, in the patient population with different CKD stages, even if there is a deviation in sampling time. However, this evaluation approach, without incorporating covariates, could be applicable only to iohexol because this drug was renally excreted. Moreover, the deviation in sampling time seems less impactful (or more beneficial) to patients with more severe CKD, as they have slower CL.  So, iohexol is an appropriate candidate drug to evaluate LSS without covariate-incorporated models, but it doesn't seem to be applicable to other drugs (e.g. mainly metabolized in the liver). Therefore, the conclusion should be limited to this drug. 

Author Response

(The authors gave the same response as above.)

Round 2

Reviewer 3 Report

NONE

Author Response

Comments from reviewers

No additional comments from the reviewers were noted. We are grateful to the reviewers for their previous feedback on our work, which we have used to further improve the manuscript.

Reviewer 4 Report

No further comments. 

Author Response

(The authors gave the same response as above.)
